# Impact of Epicardial Adipose Tissue on Infarct Size and Left Ventricular Systolic Function in Patients with Anterior ST-Segment Elevation Myocardial Infarction

**DOI:** 10.3390/diagnostics14040368

**Published:** 2024-02-07

**Authors:** Jose Gavara, Hector Merenciano-Gonzalez, Jordi Llopis-Lorente, Tamara Molina-Garcia, Nerea Perez-Solé, Elena de Dios, Víctor Marcos-Garces, Jose V. Monmeneu, Maria P. Lopez-Lereu, Joaquim Canoves, Clara Bonanad, David Moratal, Julio Núñez, Antoni Bayés-Genis, Juan Sanchis, Francisco J. Chorro, Cesar Rios-Navarro, Vicente Bodí

**Affiliations:** 1Center for Biomaterials and Tissue Engineering, Universitat Politècnica de València, 46022 Valencia, Spain; jose.gavara@outlook.es (J.G.); dmoratal@eln.upv.es (D.M.); 2Instituto de Investigación Sanitaria INCLIVA, 46010 Valencia, Spain; hectormeren@gmail.com (H.M.-G.); tamara5mg1999@gmail.com (T.M.-G.); neere_8@hotmail.com (N.P.-S.); vic_mg_cs@hotmail.com (V.M.-G.); xcfemenia@gmail.com (J.C.); clarabonanad@gmail.com (C.B.); yulnunez@gmail.com (J.N.); sanchis_juafor@gva.es (J.S.); francisco.j.chorro@uv.es (F.J.C.); 3Centro de Investigación e Innovación en Bioingeniería (Ci2B), Universitat Politècnica de València, 46010 Valencia, Spain; jorllolo@alumni.uv.es; 4Centro de Investigación Biomédica en Red—Cardiovascular (CIBER-CV), 28022 Madrid, Spain; elenaddll@gmail.com (E.d.D.); abayesgenis@gmail.com (A.B.-G.); 5Department of Cardiology, Hospital Clinico Universitario de Valencia, 46010 Valencia, Spain; 6Cardiovascular Magnetic Resonance Unit, ASCIRES Biomedical Group, 46004 Valencia, Spain; jmonmeneu@eresa.com (J.V.M.); mplopezl@ascires.com (M.P.L.-L.); 7Department of Medicine, Faculty of Medicine and Odontology, University of Valencia, 46010 Valencia, Spain; 8Cardiology Department and Heart Failure Unit, Hospital Universitari Germans Trias i Pujol, 08193 Badalona, Spain; 9Department of Medicine, Universitat Autonoma de Barcelona, 08193 Barcelona, Spain; 10Department of Pathology, Faculty of Medicine and Odontology, University of Valencia, 46010 Valencia, Spain

**Keywords:** ST-segment elevation myocardial infarction, epicardial adipose tissue, cardiovascular magnetic resonance, infarct size, left ventricular ejection fraction

## Abstract

We aimed to assess the correlation of cardiovascular magnetic resonance (CMR)-derived epicardial adipose tissue (EAT) with infarct size (IS) and residual systolic function in ST-segment elevation myocardial infarction (STEMI). We enrolled patients discharged for a first anterior reperfused STEMI submitted to undergo CMR. EAT, left ventricular (LV) ejection fraction (LVEF), and IS were quantified at the 1-week (*n* = 221) and at 6-month CMR (*n* = 167). At 1-week CMR, mean EAT was 31 ± 13 mL/m^2^. Patients with high EAT volume (*n* = 72) showed larger 1-week IS. After adjustment, EAT extent was independently related to 1-week IS. In patients with large IS at 1 week (>30% of LV mass, *n* = 88), those with high EAT showed more preserved 6-month LVEF. This association persisted after adjustment and in a 1:1 propensity score-matched patient subset. Overall, EAT decreased at 6 months. In patients with large IS, a greater reduction of EAT was associated with more preserved 6-month LVEF. In STEMI, a higher presence of EAT was associated with a larger IS. Nevertheless, in patients with large infarctions, high EAT and greater subsequent EAT reduction were linked to more preserved LVEF in the chronic phase. This dual and paradoxical effect of EAT fuels the need for further research in this field.

## 1. Introduction

ST-segment elevation myocardial infarction (STEMI) is one of the main causes of morbidity and mortality worldwide [1]. Infarct size (IS) and residual left ventricular (LV) systolic function are seemingly the most potent predictors of subsequent adverse cardiovascular events [1,2,3]. Further research is therefore needed to better understand the determinants of IS extent after reperfusion and of systolic function in the chronic phase.

Epicardial adipose tissue (EAT) is a biologically active fat depot with pleiotropic effects located between the myocardium and the visceral layer of the pericardium. Various studies have linked the extent of EAT with the occurrence of major adverse cardiovascular events in different cardiovascular scenarios [4,5]. In patients with STEMI, however, the association of EAT with IS and residual systolic function has scarcely been investigated and the results are controversial. Likewise, the dynamics of EAT within the months following infarction and its influence on these parameters are a largely unknown area.

Cardiovascular magnetic resonance (CMR) has become the gold standard non-invasive cardiac imaging technique for accurately characterizing IS and systolic function [1,2,3]. Furthermore, it permits the precise quantification of EAT [6], thus representing the ideal technique for assessing the association of EAT extent and dynamics with IS and residual systolic function in post-STEMI patients.

The aims of this study were to use sequential CMR in a large and homogeneous series of patients with reperfused anterior STEMI to investigate: (i) the relationship between the extent of EAT and IS soon after infarction (at the 1-week CMR); (ii) the association of EAT extent at the 1-week CMR with residual systolic function, as derived from LV ejection fraction (LVEF) at the 6-month CMR; and (iii) the dynamics of EAT (from 1-week to 6-month CMR) and its relationship with the 6-month LVEF.

## 2. Materials and Methods

### 2.1. Study Population

The study group was recruited from the database of a prospective ongoing registry kept in a University Hospital [2,3,7]. We selected consecutive patients discharged for a first anterior STEMI treated with primary percutaneous coronary intervention (PCI) and submitted to undergo 1-week CMR between 2010 and 2016. The final study group comprised 221 patients, of which 167 were re-studied with CMR at 6 months post-STEMI. The patient flowchart is shown in Figure 1.

Clinical and angiographic characteristics were registered in all cases at admission. Biochemical data were derived from the blood sample obtained at pre-discharge. Data were prospectively recorded and immediately included in a specific database. All patients were followed up in a STEMI-specific outpatient clinic and current management recommendations were applied [1].

Concretely, Killip classes were defined as: I (patients without any clinical sign of heart failure), II (patients with crackles or rales in the lungs, elevated jugular venous pressure, and an S3 gallop), III (patients with evident acute pulmonary edema), and IV [patients with cardiogenic shock or hypotension (systolic blood pressure < 90 mmHg) and features of low cardiac output (oliguria, cyanosis, or impaired mental status) [8].

TIMI flow grade was defined as: TIMI 0, no perfusion (no flow after the obstruction point); TIMI 1, penetration without perfusion (the contrast material passes beyond the area of obstruction, but fails to opacify the entire artery); TIMI 2, partial reperfusion (the contrast opacifies the coronary bed distal to the obstruction point, but at a rate slower than normal); and TIMI 3, complete perfusion (normal coronary flow) [9].

The Global Registry of Acute Coronary Events (GRACE) estimates the risk of death among patients with acute coronary syndrome and is calculated using age, heart rate and systolic blood pressure at admission; serum creatinine at admission; ST segment depression; Killip class; cardiac arrest at admission; and elevated myocardial necrosis markers or enzymes.

The study protocol conformed to the principles for use of human subjects outlined in the Declaration of Helsinki and was approved by the local Research Ethics Committee (registry number: 2019/160). Informed consent was obtained from all participants.

### 2.2. Acquisition and Sequences of CMR Studies

Patients were scheduled to undergo 1-week (8 (6–11) days post-STEMI) and 6-month (198 (165–227) days post-STEMI) CMR according to a previously defined and validated study protocol [2,3,7].

The reasons for exclusion are shown in Figure 1. Patients were examined with a 1.5 T unit system (Magnetom Sonata; Siemens, Erlangen, Germany) according to a previously described standard protocol [3,7].

Images were acquired by means of a phased-array body surface coil during breath-holds and were triggered by electrocardiography.

Cine images were acquired in two-, three-, and four-chamber views, as well as in short-axis views using a steady-state free precession sequence (repetition time/echo time: 2.8/1.2 ms; flip angle: 58 degrees; matrix: 256 × 300; field of view: 320 × 270 mm; slice thickness: 7 mm).

Late gadolinium enhancement (LGE) sequences were performed 10 min after administering gadolinium-based contrast using an inversion recovery steady-state free precession sequence (repetition time/echo time: 750/1.26 ms; flip angle: 45 degrees; matrix: 256 × 184; field of view: 340 × 235 mm; slice thickness: 7 mm). Inversion time was adjusted to nullify the normal myocardium.

### 2.3. CMR Indices Quantification

All CMR studies were analyzed offline by two expert local cardiologists with more than 15 years of experience in CMR imaging using customized software (QMASS MR 6.1.5, Medis, Leiden, The Netherlands). They were blinded to any clinical information.

LVEF (%), LV end-diastolic volume index (LVEDVI, mL/m^2^), LV end-systolic volume index (LVESVI, mL/m^2^), and LV mass index (g/m^2^) were calculated by means of the manual planimetry of endocardial and epicardial borders in short-axis view cine images.

LGE was regarded as the presence of signal intensity more than five standard deviations higher than a remote non-infarcted area in the same slice. The areas showing LGE were subsequently visually revised by manual planimetry. Infarct size (% of LV mass) was assessed as the percentage of LV mass showing LGE. Extensive IS was regarded as >30% of LV mass, a cut-off value previously validated for predicting subsequent events in patients with STEMI [7]. Microvascular obstruction (MVO, % of LV mass) was quantified by means of manual planimetry and defined as the percentage of LV mass showing a lack of contrast uptake in the tissue core exhibiting LGE [7]. EAT volume (mL/m^2^) was quantified at the 1-week and 6-month CMR by manually tracing the outer myocardial wall and the visceral layer of the pericardium on an end-diastolic short-axis view [10,11], from the most basal slice towards the most apical slice (Figure 2). Total EAT was calculated through the summation of the epicardial fat volume of each slice and indexed by the body surface area (mL/m^2^) [6,12]. As no cut-off values for EAT have been clearly defined, patients were classified into two groups according to the upper EAT tertile at the 1-week CMR (35 mL/m^2^): high EAT volume (third tertile, >35 mL/m^2^) and low EAT volume (first and second tertiles, ≤35 mL/m^2^). The percentage of cardiac volume represented by EAT (%) was calculated by the formula: [EAT/Total intra-pericardial volume] × 100.

The dynamics of EAT from 1-week to 6-month CMR (∆EAT, mL/m^2^) were calculated by the formula [EAT at 1-week CMR]–[EAT at 6-month CMR].

### 2.4. Variability of Measurements

Inter- and intra-observer variability in our laboratory for calculating all CMR indices used in the present study (with the exception of epicardial adipose tissue [EAT]) was <5%, as previously reported.

Inter- and intra-observer variability for calculating EAT are shown in Table 1. Inter-observer variability was determined by comparing the differences in the measurements of 20 CMR studies randomly sampled from the study group and separately quantified by the two operators. Intra-observer variability was determined by comparing the differences between two repeated measurements of these same 20 CMR studies carried out by one of the operators (with an interval of 1 month from the first to the second measurement). An intra-class correlation coefficient value greater than 0.90 indicates excellent reliability.

### 2.5. Endpoints and Follow-Up

We aimed to explore the following endpoints: (i) the association of EAT with IS at the 1-week CMR, (ii) the association of EAT (at the 1-week CMR) with LVEF at the 6-month CMR, and (iii) the association of ∆EAT (from 1-week to 6-month CMR) with LVEF at the 6-month CMR.

### 2.6. Statistical Analysis

The data were tested for normal distribution using the Kolmogorov–Smirnov test. Continuous normally distributed variables were shown as the mean ± standard deviation and compared using Student’s *t*-tests. Non-parametric variables were expressed as the median with the interquartile range and compared using the Mann–Whitney U-test. Group percentages were compared by means of a Chi-square test or Fisher’s exact test, when appropriate.

The associations of 1-week EAT with (a) 1-week IS and (b) 6-month LVEF were assessed using multivariate forward lineal regression analyses adjusted for those baseline clinical, angiographic and biochemical variables recorded upon patient admission and yielding a *p*-value < 0.05 in univariate analyses. Unstandardized beta coefficients were computed.

In patients with extensive (>30% of LV mass) and non-extensive (≤30% of LV mass) 1-week IS, the association of 1-week EAT with 6-month LVEF was performed using forward lineal regression analyses adjusted for those variables associated with 6-month LVEF in the multivariate analysis of the entire cohort (heart rate and TIMI flow grade before PCI).

The assumptions of normality of residuals (Q-Q plots), linearity and homoscedasticity (fitted vs. residuals plots) were confirmed by visual inspection of the respective plots derived from the final linear regression models used to predict the 1-week IS and 6-month LVEF. The assumption of collinearity of variables was assessed using the tolerance statistic (excessive if <0.20) and variance inflation factor (excessive if >5).

We sought to minimize a potential selection bias in the salutary effect of extensive 1-week EAT on 6-month LVEF detected in patients with large IS, whereby less depressed 6-month LVEF in patients with extensive 1-week EAT could be the result of a less altered baseline profile or less extensive 1-week IS in this subset, and not only the result of more extensive 1-week EAT. For this purpose, the effect of the extent of 1-week EAT on 6-month LVEF in patients with a large 1-week IS was specifically addressed in a 1:1 matched population. Both groups were well balanced for all baseline parameters, showing an independent association with 1-week IS and 6-month LVEF, as well as for 1-week IS. The probit model with 1:1 nearest neighbors matching and without replacement was used to identify one patient with extensive 1-week EAT for each patient with non-extensive 1-week EAT in the cohort of patients with large IS (>30% of LV mass). The standardized mean differences calculated using Yang and Dalton’s method ≤0.2 were used as a proxy of covariate balance.

Finally, the dynamics of EAT (∆EAT from 1-week to 6-month CMR) were calculated, and their association with 6-month LVEF was determined using univariate and multivariate lineal regression analysis. The lineal multivariate regression analysis was carried out separately in patients with (>30% of LV mass) and without (≤30% of LV mass) extensive 1-week IS.

In patients with extensive (>30% of LV mass) and non-extensive (≤30% of LV mass) 1-week IS, the association of ∆EAT with 6-month LVEF was performed using forward lineal regression analyses adjusted for those variables associated with 6-month LVEF in a multivariate analysis of the entire cohort (heart rate and TIMI flow grade before PCI).

All *p*-values were two-sided. Statistical significance was achieved at a two-tailed *p*-value < 0.05. The SPSS 15.0 software (SPSS Inc., Chicago, IL, USA) and STATA 9.0 (StataCorp, College Station, TX, USA) were used throughout.

## 3. Results

Baseline clinical-angiographic, biochemical–electrocardiogram, and CMR characteristics of the entire study group as well as those of patients with and without high EAT are displayed in Table 2, Table 3 and Table 4 respectively.

### 3.1. Association of 1-Week EAT with IS

At the 1-week CMR, mean EAT was 31 ± 13 mL/m^2^, which represented 13.5% of the entire intra-pericardial cardiac volume. Compared with patients with low EAT (*n* = 149), those with high EAT (*n* = 72) exhibited a worse clinical and metabolic profile: they were older; had more frequent previous history of diabetes mellitus; and displayed higher GRACE scores, glucose, glycated hemoglobin, and triglyceride values (*p*-value < 0.05 for all comparisons) (Table 2 and Table 3).

The CMR data of patients with high and low EAT at 1 week and at 6 months are shown in Table 4. The former exhibited larger IS at the 1-week (Figure 3A, Table 4) and 6-month CMR (Table 4), *p*-value < 0.05 for all comparisons.

In the lineal multivariate regression analysis, after adjustment for baseline parameters related to IS at 1 week in univariate analysis (shown in Table 5), higher EAT was independently associated with larger 1-week IS (unstandardized β coefficient = 0.177, *p*-value = 0.040) (Table 5, Figure 3B).

### 3.2. Association of 1-Week EAT with 6-Month LVEF

At 6 months, 167 patients were re-studied with CMR; the reasons for exclusion are depicted in Figure 1. Compared to the 1-week CMR, a significant improvement in LVEF was detected at 6 months: 50 ± 13% vs. 53 ± 14%, *p*-value < 0.001. The CMR data at 6 months in the entire cohort as well as in patients with and without high 1-week EAT are depicted in Table 4.

In the lineal multivariate regression analysis, once adjusted for baseline parameters related to the 6-month LVEF in univariate analysis shown in Table 6, higher EAT at the 1-week CMR was not associated with 6-month LVEF (Table 6 and Figure 4).

In the separate lineal multivariate regression analysis of patients with non-extensive IS at the 1-week CMR, 1-week EAT was not associated with 6-month LVEF (Table 7A, Figure 5A). Nevertheless, in those with extensive IS at the 1-week CMR, 1-week EAT displayed an independent and direct association with 6-month LVEF (unstandardized β coefficient = 0.224, *p*-value = 0.037) (Table 7B, Figure 5B).

In terms of 6-month LVEF, when patients with high and low EAT were compared, no significant differences were detected in the subgroup with non-extensive 1-week IS (Figure 5C). Nevertheless, in patients with extensive IS at the 1-week CMR, those with high 1-week EAT exhibited more preserved LVEF at 6 months (Figure 5D).

To specifically analyze the association between the extent of 1-week EAT and 6-month LVEF in patients with large 1-week IS, we addressed this issue separately by selecting a 1:1 propensity score-matched population (*n* = 56): one patient with a large 1-week IS and large 1-week EAT (*n* = 28) for each patient with a large 1-week IS and low 1-week EAT (*n* = 28). As shown in Table 8, both groups were well balanced in terms of 1-week IS and baseline parameters independently related to IS (Table 5) and 6-month LVEF (Table 6). The CMR data of the matched population are also shown in Table 8. Compared with matched patients with low EAT, those with high EAT displayed more preserved 6-month LVEF (Table 8). Although the 6-month LVEF (compared with 1-week LVEF) did not vary in patients with low EAT (40 ± 10% vs. 40 ± 11%, *p*-value = 0.450), a significant improvement occurred in those with high EAT (42 ± 10% vs. 46 ± 9%, *p*-value = 0.046) (Figure 6).

### 3.3. Dynamics of EAT and Association with 6-Month LVEF

EAT diminished from the 1-week to 6-month CMR (31 ± 13 vs. 26 ± 8 mL/m^2^, *p*-value < 0.001) (Figure 7). Overall, the median ∆EAT [interquartile range] was −4 [−12–5] mL/m^2^. Compared with patients with non-extensive IS, a bigger EAT reduction occurred in those with extensive 1-week IS: −8 [−21–−2] mL/m^2^ vs. −2 [−9–8] mL/m^2^, *p*-value < 0.001 (Figure 7).

In the lineal multivariate regression analysis of the entire cohort, after adjusting for the baseline parameters independently related to 6-month LVEF shown in Table 6, ∆EAT did not correlate with 6-month LVEF (Table 9).

In the separate lineal multivariate regression analysis of patients with non-extensive 1-week IS, ∆EAT did not correlate with 6-month LVEF (Table 10A, Figure 8A). Nevertheless, in those with extensive 1-week IS, ∆EAT displayed an independent association with 6-month LVEF (unstandardized β coefficient = −0.347, *p*-value = 0.003) (Table 10B, Figure 8B).

In patients with extensive 1-week IS, those with an EAT decrease at the 6-month CMR (∆EAT <0 mL/m^2^, first tertile) displayed more preserved 6-month LVEF than those with ∆EAT ≥0 mL/m^2^ (*p*-value = 0.026, Figure 9). These differences were not observed in patients with non-extensive IS.

## 4. Discussion

The present study shows that in a homogeneous series of patients with a first anterior STEMI reperfused with a percutaneous intervention and sequentially studied with CMR, an elevated presence of EAT is associated with more extensive IS. In patients with large infarctions, however, high EAT and greater subsequent EAT reduction correlates with more preserved LVEF in the chronic phase (Figure 10). In the specific post-infarction setting, this dual and paradoxical influence of EAT somehow resembles the obesity paradox and its yin–yang effects in the field of cardiovascular diseases [13].

### 4.1. Anatomy and Physiology of EAT

EAT is a biologically active fat depot representing a major component of the cardiac structure [4,5,6], and which occupied 13.5% of the entire intra-pericardial cardiac volume in the present study. Microscopically, EAT is composed mainly of adipocytes, although it also contains nerves and a rich variety of inflammatory cells and fibroblasts. Histologically, it is considered white adipose tissue, although it also has characteristics in common with brown and beige adipose tissues [14]. EAT and the myocardium share the same microcirculation, and this implies close structural and metabolic interaction [5]. Indeed, both salutary and potentially deleterious EAT-mediated effects on the myocardium have been suggested that seem partly associated with the pleiotropic simultaneous pro- and anti-inflammatory characteristics of this tissue [4,5,6,14]. However, data are lacking on the effect of EAT volume (and its dynamics) on the most relevant prognostic markers post-STEMI, namely IS and residual systolic function.

### 4.2. EAT and IS

The present study represents the largest series so far to assess the impact of EAT on IS as derived from CMR. Patients with high EAT were found to display a larger infarcted area than those with low EAT, and after adjustment we detected an independent and direct association between EAT and IS.

The relationship between pre-existing EAT at the time of acute coronary occlusion and resulting IS has been controversial. Concurring with our results, Islas et al. [15], Mohamed et al. [16], Fisser et al. [11], and Toya et al. [17]. reported larger infarctions in patients with high EAT; however, Bière et al. [18] and Gohbara et al. [12] detected the opposite. Differences in methodology (some studies used echocardiography instead of CMR or computed tomography for quantification) and the analysis of heterogeneous study groups (including mixed populations with both anterior and inferior infarctions or ST- and non-ST-elevation acute coronary syndromes) might account for these discrepancies.

To avoid these potentially confounding factors, we selected a large homogeneous series made of patients with a first reperfused anterior STEMI studied with CMR, the current gold-standard technique for a simultaneous quantification of IS [1] and EAT [6]. The association of EAT with IS detected at the 1-week CMR can be at least partly explained by the following factors:(i)Patients with high EAT were older; had more frequent diabetes mellitus; and displayed higher GRACE scores, glucose, glycated hemoglobin, and triglyceride values. The association of EAT with a worse clinical and metabolic profile is not new and probably contributed to a more deteriorated coronary circulation that resulted in larger infarctions in patients with extensive EAT. In a short series of 54 patients with STEMI studied with CMR, Fisser et al. [11] reported that MVO was more frequent in those with high EAT. Indeed, in our series, these cases displayed a higher Bypass Angioplasty Revascularization Investigation score (reflecting a more severe coronary atherosclerotic burden) and more extensive MVO (due to less efficient microvascular reperfusion). Both factors could eventually contribute to the more severe structural damage detected in patients with high EAT.(ii)EAT is a biologically active tissue closely connected with the myocardium [5]. Parallel to the response detected in the myocardium itself, in the acute post-infarction period EAT can also undergo a deregulated pro-inflammatory response which can ultimately contribute to an undesirable expansion of structural damage [4,5,6,15].

In summary, our data strongly suggest that whether due to an unfavorable pre-existing metabolic and coronary profile or deleterious biological effects in the acute phase, pre-existing high EAT is linked to a more extensive IS.

### 4.3. Association of EAT with 6-Month LVEF

It has been widely demonstrated that residual LVEF in the chronic phase is the most robust long-term risk predictor post-STEMI [1,3]. Based on the direct association of EAT with IS at the 1-week CMR and the fact that 1-week IS was the most potent predictor of 6-month LVEF in multivariate analyses, it could be speculated that higher EAT at 1 week would be related to more depressed LV systolic function and more LV remodeling in the chronic phase. Nevertheless, a more detailed examination of our results in patients with extensive 1-week IS revealed that the larger the extent of 1-week EAT, the more preserved the 6-month LVEF and the lesser the 6-month LV dilation. After adjustment, and in a propensity score-matched population, these salutary associations of 1-week EAT with 6-month LVEF and LV remodeling existed in patients with large 1-week IS (those who in general tend to present more severe LV systolic deterioration) but not in those with non-extensive 1-week IS (in whom systolic function is preserved or almost completely preserved since the acute phase).

The implications of EAT on late systolic function and LV remodeling are somewhat understudied and the results are controversial. Bière et al. [18] found no significant association of EAT with 3-month LVEF; however, they did not undertake a separate analysis in patients with large infarctions. Nevertheless, in the context of heart failure, Pugliese et al. [19] obtained results similar to ours: EAT exerted salutary structural and prognostic effects in patients with heart failure and reduced LVEF, but not in those with preserved LVEF. The authors suggested that in patients with depressed systolic function, EAT might act as a metabolic reservoir that would ultimately exert protective effects on an already deteriorated myocardium.

Although beyond the scope of this study, it could be speculated that several pathophysiological mechanisms underlie the beneficial structural effects of EAT in the long term. First, the predominance of an anti-inflammatory response and the release of cardioprotective cytokines such as adiponectin [20] or pregnancy-associated plasma protein-A [21] in the chronic phase after infarction by EAT could favorably stimulate the neighboring myocardium towards more effective reparation and systolic recovery. Second, EAT-derived nutrients could act as an energy reservoir, thereby enabling viable cardiomyocytes to contract more efficiently [19]. Third, the activation of the thermogenic pathway in the brown adipose tissue through uncoupling protein-1 expression has been shown to exert a beneficial effect on myocardial performance [22]. Finally, larger amounts of EAT soon after infarction could serve to attenuate the natural tendency towards subsequent LV dilatation in patients with large infarctions, purely through physical containment.

### 4.4. EAT Dynamics

A significant trend towards decreased EAT volume was observed within the first 6 months post-STEMI. Both intensive lipid-lowering therapy and changes in dietary and exercise habits likely favor the reduction of adipose tissue; however, EAT consumption by the myocardium itself may also play a key role. In fact, Chang at al. [23] showed that EAT lipolysis increased significantly in infarcted rats compared to controls.

Interestingly, this tendency was more pronounced in patients with extensive infarction. In this subset, more substantial EAT reduction was associated with more preserved LVEF and less dilated LV volumes 6 months post-STEMI. These results are consistent with those of a previous study that used echocardiography to evaluate a small sample size. In a cohort of 100 patients with STEMI, Parisi et al. reported that EAT reduction was linked to less dilated LV end-diastolic volume and more preserved LVEF, proposing that this finding could be mediated by the release of anti-inflammatory interleukin-13 [24].

One hypothesis regarding the chronic phase post-STEMI is that EAT acts as a reservoir of energy and protective products, whose consumption favors the repair of the neighboring infarcted myocardium. This could explain why EAT reduction is more notable and the benefit of this reduction is more significant in patients with a greater demand, namely those with larger infarctions. However, this theory is only for hypothesis-generating purposes and should be further analyzed and confirmed in dedicated subsequent studies.

### 4.5. Study Limitations

The present study is purely observational and strictly reflects the structural implications of EAT volume and its dynamics in patients with STEMI as derived from CMR. The sample size is insufficient to evaluate the prognostic value of 1-week EAT in the post-STEMI scenario. The underlying pathophysiological mechanisms and potential therapeutic implications of these findings are beyond the scope of our study.

## 5. Conclusions

In summary, our observations in the specific post-STEMI scenario resemble the previously described obesity paradox. This refers to the apparent contradictory fact that on the one hand, obesity represents a risk factor for cardiovascular diseases, whereas, on the other hand, already established cardiovascular diseases tend to follow a more benign course in obese patients. Similarly, pre-existing extensive EAT in patients with STEMI in-creases the probability of severe structural damage in the acute phase. Nevertheless, in patients with an already established large myocardial infarction, the presence of pre-existing high EAT, as well as the consumption of this tissue, may protect against subsequent systolic function deterioration and LV dilation. More clinical and basic research is needed to confirm this dual role of EAT after STEMI and further elucidate the underlying mechanisms, as well as the potential therapeutic implications.

## Figures and Tables

**Figure 1 diagnostics-14-00368-f001:**
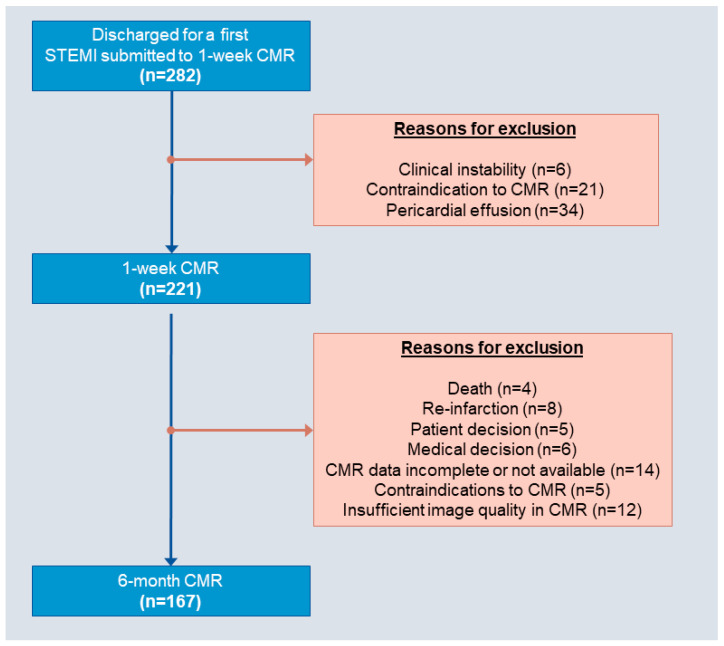
Flowchart of patients. Reasons for exclusion are depicted in boxes. Abbreviations: CMR = cardiovascular magnetic resonance; STEMI = ST-segment elevation myocardial infarction.

**Figure 2 diagnostics-14-00368-f002:**
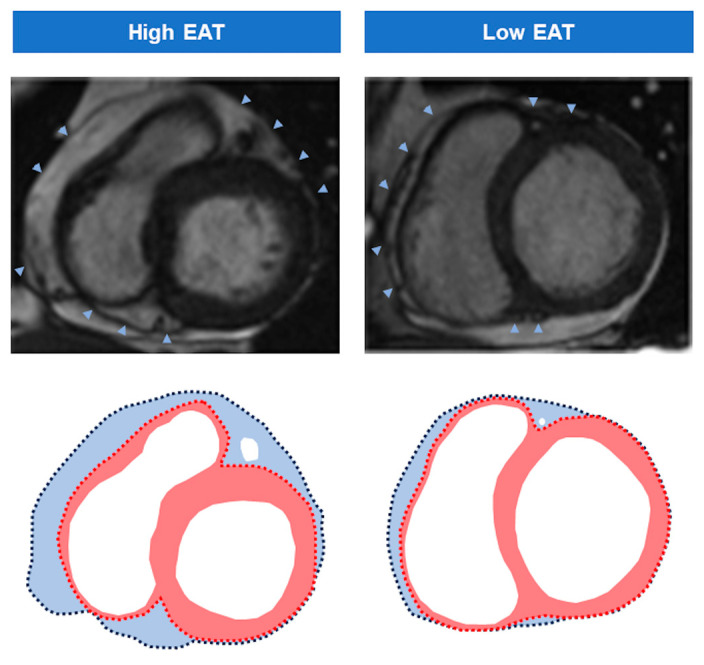
EAT measurement. EAT (light blue area) was quantified at the 1-week and 6-month cardiovascular magnetic resonance by manually tracing the outer myocardial wall (red dotted line) and the visceral layer of the pericardium (arrowheads and dark blue dotted line). The left panel illustrates a patient with high EAT (>third tertile, 35 mL/m^2^) and the right panel corresponds to a patient with low EAT (≤35 mL/m^2^). Abbreviation: EAT = Epicardial adipose tissue.

**Figure 3 diagnostics-14-00368-f003:**
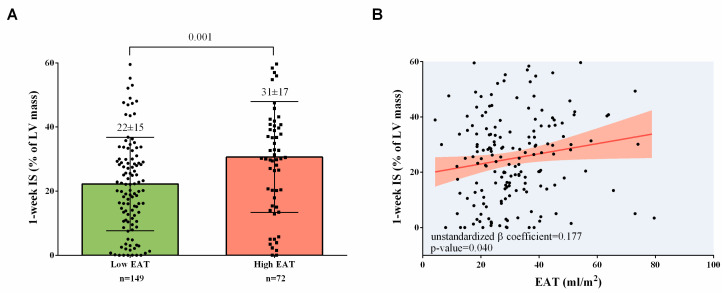
Association of EAT with IS at 1-week CMR. (**A**) Patients with high EAT (>35 mL/m^2^) displayed larger IS at 1-week CMR. (**B**) After adjustment, EAT showed a significant and direct association with IS. Abbreviations: CMR = Cardiovascular magnetic resonance. EAT = Epicardial adipose tissue. IS = Infarct size. LV = Left ventricular.

**Figure 4 diagnostics-14-00368-f004:**
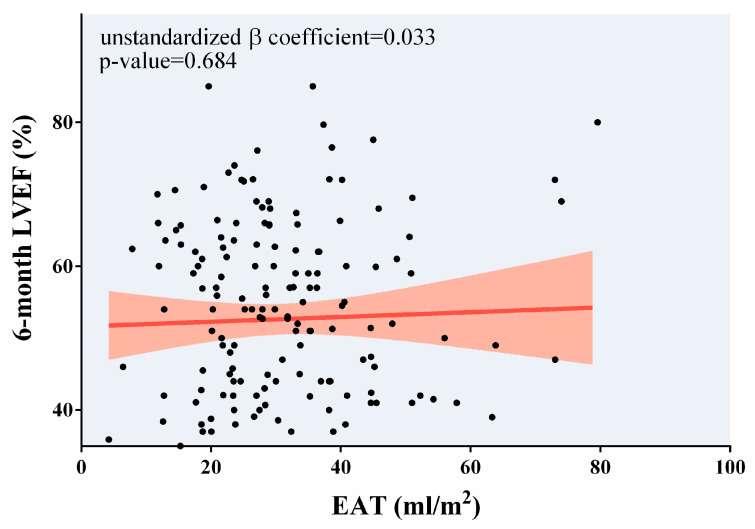
Association of 1-week EAT with 6-month LVEF. After adjustment, 1-week EAT did not show a significant association with 6-month LVEF. Abbreviations: EAT = Epicardial adipose tissue. LVEF = Left ventricular ejection fraction.

**Figure 5 diagnostics-14-00368-f005:**
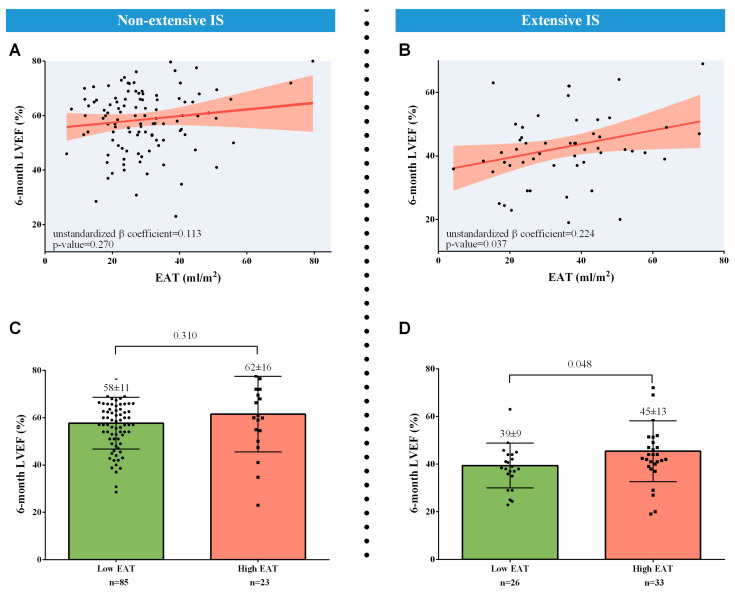
Association of 1-week EAT with 6-month LVEF. After adjustment, 1-week EAT was not correlated with 6-month LVEF in patients with non-extensive IS (**A**). Nevertheless, an independent and direct association of 1-week EAT with 6-month LVEF existed in those with extensive IS at the 1-week cardiovascular magnetic resonance (**B**). No significant differences between patients with high and low 1-week EAT were detected in terms of 6-month LVEF in the subgroup with non-extensive 1-week IS (**C**). Nevertheless, in patients with extensive 1-week IS, those with high 1-week EAT exhibited more preserved LVEF at 6 months (**D**). Abbreviations: EAT = Epicardial adipose tissue. IS = Infarct size. LVEF = Left ventricular ejection fraction.

**Figure 6 diagnostics-14-00368-f006:**
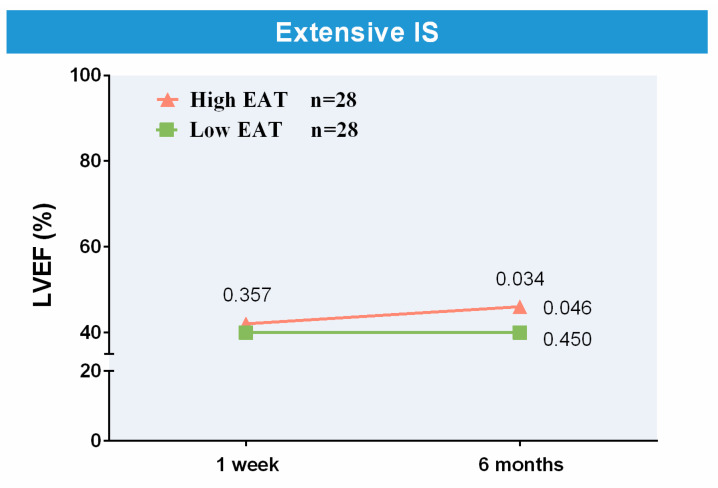
Association of 1-week EAT with LVEF in the matched population. In the matched patients, compared with 1-week LVEF, 6-month LVEF improved in those with high EAT but did not vary in patients with low 1-week EAT. Abbreviations. EAT = Epicardial adipose tissue. IS = Infarct size. LVEF = Left ventricular ejection fraction.

**Figure 7 diagnostics-14-00368-f007:**
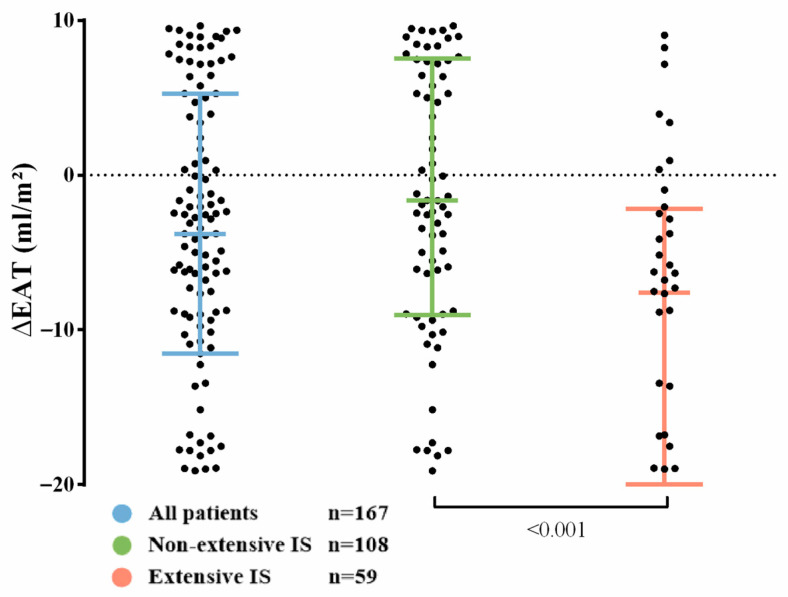
Dynamics of EAT. EAT decreased from the 1-week to the 6-month cardiovascular magnetic resonance. Compared with patients with non-extensive IS, more substantial EAT reduction occurred in those with extensive 1-week IS. Abbreviations: CMR = Cardiovascular magnetic resonance. EAT = Epicardial adipose tissue. IS = Infarct size. ∆EAT = Variation in EAT (mL/m^2^) from 1-week to 6-month CMR.

**Figure 8 diagnostics-14-00368-f008:**
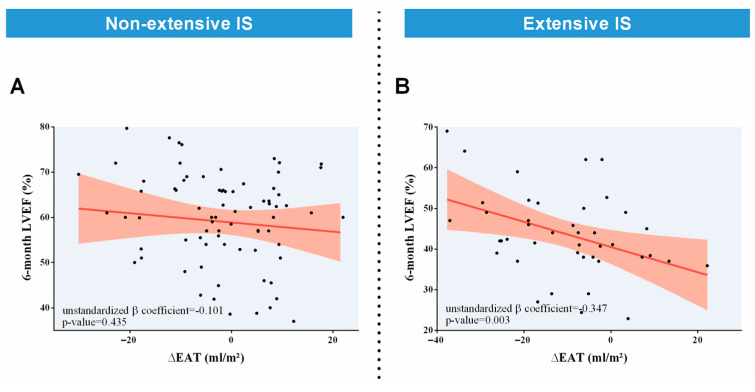
Association of ∆EAT with 6-month LVEF. ∆EAT was not associated with 6-month LVEF in patients with non-extensive IS (**A**). Nevertheless, an independent association of ∆EAT with 6-month LVEF existed in those with extensive 1-week IS (**B**). The *p*-value for the interaction between non-extensive and extensive IS was <0.05. Abbreviations. CMR = Cardiovascular magnetic resonance. EAT = Epicardial adipose tissue. IS = Infarct size. LVEF = Left ventricular ejection fraction. ∆EAT = Variation of EAT (mL/m^2^) from 1-week to 6-month CMR.

**Figure 9 diagnostics-14-00368-f009:**
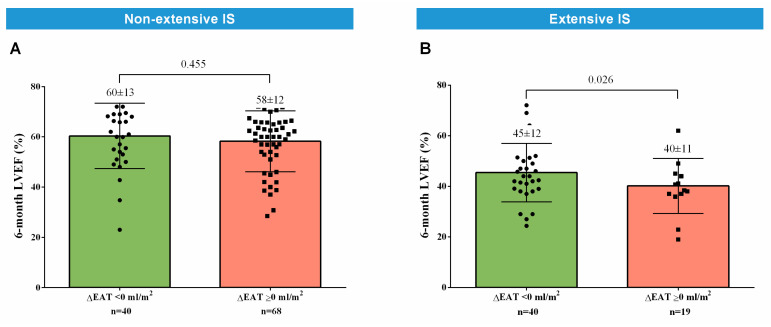
Association of ∆EAT with 6-month LVEF. In patients with extensive 1-week IS, those with ∆EAT < 0 mL/m^2^ (first tertile) displayed more preserved 6-month LVEF than those with ∆EAT ≥ 0 mL/m^2^ (**A**). These differences did not occur in patient with non-extensive IS (**B**). Abbreviations: CMR = Cardiovascular magnetic resonance. EAT = Epicardial adipose. IS = Infarct size. LVEF = Left ventricular ejection fraction. ∆EAT = Variation of EAT (mL/m^2^) from 1-week to 6-month CMR.

**Figure 10 diagnostics-14-00368-f010:**
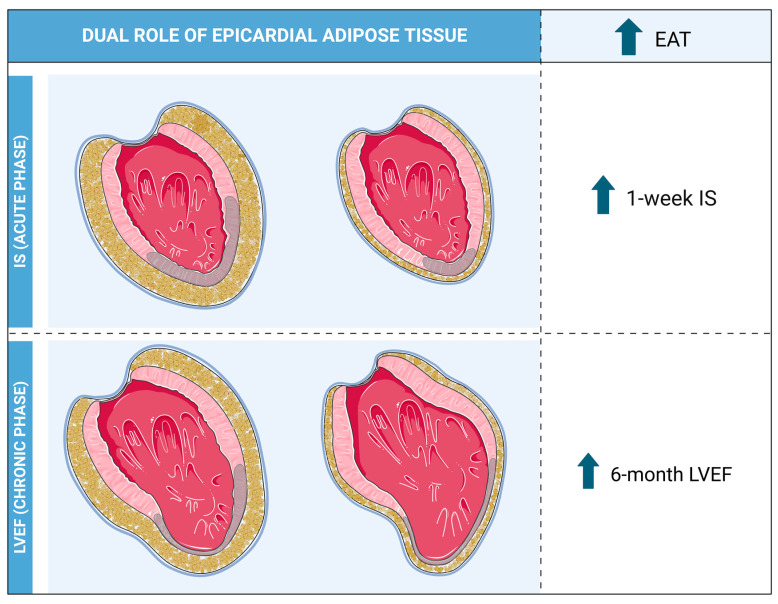
Central Illustration. In patients with a first anterior STEMI, the presence of high EAT upon reperfusion is associated with the occurrence of large infarctions (upper panels). However, in patients with already established large infarctions, high EAT and greater subsequent EAT reduction are related to more preserved left ventricular systolic function in the chronic phase (lower panels). Abbreviations: EAT = Epicardial adipose tissue. IS = Infarct size. LVEF = Left ventricular ejection fraction. STEMI = ST-segment elevation myocardial infarction. ∆EAT = Variation of EAT from 1-week to 6-month cardiovascular magnetic resonance.

**Table 1 diagnostics-14-00368-t001:** Variability in EAT measurements.

	Relative Change	Absolute Change	Intra-Class Correlation Coefficient
** * Inter-observer variability * **			
**EAT (mL/m^2^)**	6.4 ± 10.8%	1.9 ± 2.9 mL/m^2^	0.959
** * Intra-observer variability * **			
**EAT (mL/m^2^)**	3.7 ± 4.4%	0.8 ± 1.0 mL/m^2^	0.996

Abbreviation. EAT = Epicardial adipose tissue.

**Table 2 diagnostics-14-00368-t002:** Baseline clinical and angiographic characteristics and therapies of the entire study group and of patients with high and low EAT.

	All Patients	EAT ≤ 35 mL/m^2^	EAT > 35 mL/m^2^	*p*-Value
**Number of patients**	221	149	72	
** * Baseline characteristics * **				
**Age (years)**	58 ± 13	56 ± 13	61 ± 13	0.016
**Male sex (%)**	180 (81)	122 (82)	58 (81)	0.812
**Diabetes mellitus (%)**	45 (20)	22 (15)	23 (32)	0.003
**Hypertension (%)**	89 (40)	55 (37)	34 (47)	0.143
**Hypercholesterolemia (%)**	93 (42)	59 (40)	34 (47)	0.282
**Smoker (%)**	137 (62)	91 (61)	46 (64)	0.508
**Body mass index (kg/m^2^)**	27 ± 4	27 ± 5	27 ± 3	0.909
**Heart rate (bpm)**	83 ± 21	83 ± 22	82 ± 18	0.674
**Systolic blood pressure (mmHg)**	134 ± 29	134 ± 27	133 ± 32	0.797
**Killip class (%)**				0.401
** 1**	182 (82)	125 (84)	57 (79)	
** 2**	29 (13)	17 (11)	12 (17)	
** 3**	3 (1)	3 (2)	0 (0)	
** 4**	7 (3)	4 (3)	3 (4)	
**Time to reperfusion (min)**	201 [127–348]	190 [120–334]	208 [130–407]	0.474
**Proximal LAD disease (%)**	104 (47)	71 (48)	33 (46)	0.628
**Multivessel disease (%)**	67 (30)	43 (29)	24 (33)	0.515
**BARI score**	38 [25–50]	38 [25–50]	38 [25–50]	0.558
**TIMI flow grade before PCI (%)**				0.975
** 0**	115 (52)	76 (51)	39 (54)	
** 1**	9 (4)	6 (4)	3 (4)	
** 2**	24 (11)	17 (11)	7 (10)	
** 3**	73 (33)	50 (34)	23 (32)	
**TIMI flow grade after PCI (%)**				0.039
** 0**	4 (2)	4 (2)	0 (0)	
** 1**	1 (0)	1 (1)	0 (0)	
** 2**	17 (8)	7 (5)	10 (14)	
** 3**	199 (90)	137 (92)	62 (86)	
**GRACE score**	128 ± 33	124 ± 31	134 ± 36	0.032
** * Medical treatment at discharge * **				
**DAPT (%)**	205 (93)	137 (92)	68 (94)	0.502
**Statins (%)**	193 (87)	128 (86)	65 (90)	0.360
**High-dose statins * (%)**	129 (58)	87 (58)	42 (58)	0.994
**ACEi/ARA/ARNi (%)**	169 (77)	113 (76)	56 (78)	0.750
**Beta-blockers (%)**	182 (82)	123 (83)	59 (82)	0.912
**Diuretics (%)**	29 (13)	19 (13)	10 (14)	0.814
**Anticoagulants (%)**	29 (13)	18 (12)	11 (15)	0.509
**Mineralocorticoid ** **receptor antagonists (%)**	36 (21)	24 (21)	12 (22)	0.932

Abbreviations: ACEi = Angiotensin-converting enzyme inhibitors. ARA = Angiotensin II receptor antagonists. ARNi = Angiotensin receptor neprilysin inhibitors. BARI = Bypass Angioplasty Revascularization Investigation. DAPT = Dual antiplatelet therapy. EAT = Epicardial adipose tissue. GRACE = Global Registry of Acute Coronary Events. LAD = Left anterior descending. PCI = Percutaneous coronary intervention. TIMI = Thrombolysis in Myocardial Infarction. * High dose statin indicates atorvastatin ≥ 40 mg daily or rosuvastatin ≥ 20 mg daily.

**Table 3 diagnostics-14-00368-t003:** Biochemical and ECG characteristics of the entire study group and of patients with high and low EAT.

	All Patients	EAT ≤ 35 mL/m^2^	EAT > 35 mL/m^2^	*p*-Value
**Number of patients**	221	149	72	
** * Biochemical characteristics * **				
**Glucose (mg/dL)**	105 [96–125]	103 [94–121]	114 [101–135]	0.003
**HbA1c (%)**	5.6 [5.4–6]	5.6 [5.3–5.9]	5.8 [5.5–6.9]	0.002
**Creatinine (mg/dL)**	0.9 [0.8–1.1]	0.9 [0.8–1.1]	1.0 [0.8–1.1]	0.164
**Uric acid (mg/dL)**	5.7 ± 1.8	5.7 ± 1.8	5.7 ± 1.7	0.823
**Total cholesterol (mg/dL)**	186 ± 48	182 ± 46	193 ± 52	0.116
**LDL cholesterol (mg/dL)**	121 ± 39	118 ± 36	126 ± 44	0.183
**HDL cholesterol (mg/dL)**	41 ± 11	42 ± 12	40 ± 10	0.385
**Triglycerides (mg/dL)**	126 [99–168]	121 [94–161]	144 [111–180]	0.002
**ECG characteristics**				
**Q-leads**	3 [2–4]	3 [2–4]	3 [1–4]	0.559
**ST-segment resolution (%)**	71 [50–86]	71 [43–86]	73 [50–86]	0.337

Abbreviations: EAT = Epicardial adipose tissue. ECG = electrocardiogram. HbA1c = Glycated hemoglobin. HDL = High-density lipoprotein. LDL = Low-density lipoprotein.

**Table 4 diagnostics-14-00368-t004:** CMR characteristics of the entire study group and of patients with high and low EAT.

	All Patients	EAT ≤ 35 mL/m^2^	EAT > 35 mL/m^2^	*p*-Value
** * 1-week CMR * **				
**Number of patients**	221	149	72	
**LVEF (%)**	50 ± 13	51 ± 12	49 ± 13	0.248
**LVEDVI (mL/m^2^)**	80 ± 22	81 ± 22	77 ± 22	0.206
**LVESVI (mL/m^2^)**	41 ± 19	41 ± 19	41 ± 19	0.935
**LV mass (g/m^2^)**	77 ± 19	76 ± 17	79 ± 22	0.173
**MVO (% of LV mass)**	0 [0–2.8]	0 [0–2.2]	1.1 [0–3.9]	0.081
**IS (% of LV mass)**	25 ± 16	22 ± 15	31 ± 17	0.001

** * 6-month CMR * **				
**Number of patients**	167	113	54	
**LVEF (%)**	53 ± 14	53 ± 13	52 ± 16	0.640
**LVEDVI (mL/m^2^)**	82 ± 28	83 ± 28	81 ± 28	0.650
**LVESVI (mL/m^2^)**	41 ± 25	41 ± 24	41 ± 26	0.946
**LV mass (g/m^2^)**	70 ± 19	69 ± 19	73 ± 18	0.137
**MVO (% of LV mass)**	0 [0,0]	0 [0,0]	0 [0,0]	0.095
**IS (% of LV mass)**	22 ± 14	20 ± 13	27 ± 15	0.007

Abbreviations: CMR = Cardiovascular magnetic resonance. EAT = Epicardial adipose tissue. IS = Infarct size. LV = Left ventricular. LVEDVI = Left ventricular end-diastolic volume index. LVEF = Left ventricular ejection fraction. LVESVI = Left ventricular end-systolic volume index. MVO = Microvascular obstruction.

**Table 5 diagnostics-14-00368-t005:** Association of baseline characteristics and EAT with 1-week IS; univariate and multivariate lineal regression analyses.

	Univariate		Multivariate	
	Unstandardized β Coefficient	*p*-Value	Unstandardized β Coefficient	*p*-Value
** * Baseline characteristics * **				
**Age (years)**	−0.186	0.050 *	−0.132	0.078
**Male sex (%)**	4.900	0.115	–	–
**Diabetes mellitus (%)**	2.264	0.477	–	–
**Hypertension (%)**	−2.061	0.406	–	–
**Hypercholesterolemia (%)**	−2.053	0.405	–	–
**Smoker (%)**	0.413	0.873	–	–
**Body mass index (kg/m^2^)**	−0.125	0.747	–	–
**Heart rate (bpm)**	0.173	0.003 *	0.197	<0.001
**Systolic blood pressure (mmHg)**	−0.093	0.032 *	−0.139	0.056
**Killip class (%)**	0.354	0.860	–	–
**Time to reperfusion (min)**	0.015	0.058	–	–
**Proximal LAD disease (%)**	5.273	0.033 *	5.534	0.020
**Multivessel disease (%)**	−2.495	0.362	–	–
**BARI score**	0.164	0.073	–	–
**TIMI flow grade before PCI (%)**	−1.809	0.040 *	−1.714	0.044
**TIMI flow grade after PCI (%)**	−1.169	0.639	–	–
**GRACE score**	0.027	0.498	–	–
** * Biochemical characteristics * **				
**Glucose (mg/dL)**	−0.008	0.856	–	–
**HbA1c (%)**	1.164	0.528	–	–
**Creatinine (mg/dL)**	−1.012	0.808	–	–
**Uric acid (mg/dL)**	−0.570	0.500	–	–
**Total cholesterol (mg/dL)**	−0.025	0.320	–	–
**LDL cholesterol (mg/dL)**	−0.026	0.408	–	–
**HDL cholesterol (mg/dL)**	−0.171	0.120	–	–
**Triglycerides (mg/dL)**	−0.012	0.309	–	–

**EAT (mL/m^2^)**	0.183	0.041 *	0.177	0.040

Abbreviations: BARI = Bypass Angioplasty Revascularization Investigation. EAT = Epicardial adipose tissue. GRACE = Global Registry of Acute Coronary Events. HbA1c = Glycated hemoglobin. HDL = High density lipoprotein. IS = Infarct size. LAD = Left anterior descending. LDL = Low-density lipoprotein. PCI = Percutaneous coronary intervention. TIMI = Thrombolysis in Myocardial Infarction. * Variables with *p*-value < 0.05 in the univariate analysis were tested in the multivariate analysis.

**Table 6 diagnostics-14-00368-t006:** Association of baseline characteristics and EAT with 6-month LVEF. Univariate and multivariate lineal regression analyses.

	Univariate		Multivariate	
	Unstandardized β Coefficient	*p*-Value	Unstandardized β Coefficient	*p*-Value
** * Baseline characteristics * **				
**Age (years)**	0.155	0.078	–	–
**Male sex (%)**	−4.321	0.122	–	–
**Diabetes mellitus (%)**	−1.783	0.497	–	–
**Hypertension (%)**	0.704	0.746	–	–
**Hypercholesterolemia (%)**	−0.636	0.770	–	–
**Smoker (%)**	−0.150	0.947	–	–
**Body mass index (kg/m^2^)**	0.064	0.831	–	–
**Heart rate (bpm)**	−0.178	<0.001 *	−0.215	0.003
**Systolic blood pressure (mmHg)**	0.041	0.315	–	–
**Killip class (%)**	−3.501	0.079	–	–
**Time to reperfusion (min)**	−0.005	0.029*	−0.188	0.111
**Proximal LAD disease (%)**	−3.721	0.089	–	–
**Multivessel disease (%)**	−0.023	0.992	–	–
**BARI score**	−0.138	0.109	–	–
**TIMI flow grade before PCI (%)**	1.914	0.014 *	2.641	0.020
**TIMI flow grade after PCI (%)**	5.308	0.015 *	0.046	0.706
**GRACE score**	−0.024	0.509	–	–
** * Biochemical characteristics * **				
**Glucose (mg/dL)**	−0.052	0.106	–	–
**HbA1c (%)**	−2.500	0.038 *	−0.092	0.432
**Creatinine (mg/dL)**	0.230	0.953	–	–
**Uric acid (mg/dL)**	0.466	0.534	–	–
**Total cholesterol (mg/dL)**	0.016	0.505	–	–
**LDL cholesterol (mg/dL)**	0.020	0.508	–	–
**HDL cholesterol (mg/dL)**	0.101	0.315	–	–
**Triglycerides (mg/dL)**	−0.003	0.766	–	–

**EAT (mL/m^2^)**	0.033	0.684	–	–

Abbreviations: BARI = Bypass Angioplasty Revascularization Investigation. EAT = Epicardial adipose tissue. GRACE = Global Registry of Acute Coronary Events. HbA1c = Glycated hemoglobin. HDL = High density lipoprotein. IS = Infarct size. LAD = Left anterior descending. LDL = Low-density lipoprotein. LVEF = Left ventricular ejection fraction. PCI = Percutaneous coronary intervention. TIMI = Thrombolysis in Myocardial Infarction. * Variables with *p*-value < 0.05 in the univariate analysis were tested in the multivariate analysis.

**Table 7 diagnostics-14-00368-t007:** Multivariate lineal regression analyses adjusted for parameters independently related to 6-month LVEF shown in Table 6 in patients with non-extensive (**A**) and extensive (**B**) 1-week IS.

**A. *NON-EXTENSIVE IS***				
	**Univariate**		**Multivariate**	
	**Unstandardized ** **β Coefficient**	** *p* ** **-value**	**Unstandardized ** **β Coefficient**	** *p* ** **-Value**
**Heart rate (bpm)**	−0.046	0.471	−0.045	0.486
**TIMI flow grade before PCI (%)**	0.668	0.464	0.730	0.425
**EAT (mL/m^2^)**	0.117	0.245	0.113	0.270
**B. *EXTENSIVE IS***				
	**Univariate**		**Multivariate**	
	**Unstandardized ** **β Coefficient**	** *p* ** **-value**	**Unstandardized ** **β Coefficient**	** *p* ** **-Value**
**Heart rate (bpm)**	−0.134	0.049	−0.264	0.051
**TIMI flow grade before PCI (%)**	1.724	0.135	0.181	0.187
**EAT (mL/m^2^)**	0.213	0.041	0.224	0.037

Abbreviations: EAT = Epicardial adipose tissue. IS = Infarct size. LVEF = Left ventricular ejection fraction. PCI = Percutaneous coronary intervention. TIMI = Thrombolysis in Myocardial Infarction.

**Table 8 diagnostics-14-00368-t008:** (**A**) Baseline characteristics of the 1:1 matched population (all patients and patients with high and low EAT); variables showing an independent association with 1-week IS (Table 5) and 6-month LVEF (Table 6) as well as 1-week IS were used for selecting the 1:1 matched patients. (**B**) CMR characteristics of the 1:1 matched population (all patients and patients with high and low EAT).

**A. *BASELINE CHARACTERISTICS USED FOR SELECTING THE 1:1 MATCHED PATIENTS***				
	**All patients**	**EAT ≤ 35 mL/m^2^**	**EAT > 35 mL/m^2^**	**SMD**
**Number of patients**	56	28	28	
**Heart rate (bpm)**	89 ± 22	92 ± 25	87 ± 19	0.198
**Proximal LAD disease (%)**	35 (63)	18 (64)	17 (61)	0.187
**TIMI flow grade before PCI (%)**				−0.176
**0**	36 (64)	18 (64)	18 (64)	
**1**	1 (2)	1 (4)	0 (0)	
**2**	4 (7)	2 (7)	2 (7)	
**3**	15 (27)	7 (25)	8 (29)	
**IS (% of LV mass)**	41 ± 9	41 ± 8	42 ± 10	−0.109
**B. *CMR CHARACTERISTICS***				
	**All patients**	**EAT ≤ 35 mL/m^2^**	**EAT > 35 mL/m^2^**	** *p* ** **-value**
** *CMR indices at 1 week* ** **Number of patients**	56	28	28	
**LVEF (%)**	41 ± 10	40 ± 10	42 ± 10	0.357
**LVEDVI (mL/m^2^)**	89 ± 29	96 ± 31	82 ± 27	0.083
**LVESVI (mL/m^2^)**	54 ± 25	59 ± 27	49 ± 21	0.112
**LV mass (g/m^2^)**	81 ± 15	80 ± 14	81 ± 17	0.840
**MVO (% of LV mass)**	2.3 [0.8–8.3]	2.3 [0–7.2]	2.2 [0.9–8.8]	0.873
**IS (% of LV mass)**	41 ± 9	41 ± 8	42 ± 10	0.726
** *CMR indices at 6 months* ** **Number of patients**	56	28	28	
**LVEF (%)**	44 ± 10	40 ± 11	46 ± 9	0.034
**LVEDVI (mL/m^2^)**	97 ± 35	111 ± 41	86 ± 26	0.016
**LVESVI (mL/m^2^)**	57 ± 30	72 ± 34	46 ± 21	0.003
**LV mass (g/m^2^)**	75 ± 18	76 ± 21	74 ± 16	0.694
**MVO (% of LV mass)**	0 [0–0.9]	0 [0–0.9]	0 [0–0.5]	0.435
**IS (% of LV mass)**	35 ± 7	35 ± 7	35 ± 7	0.777

Abbreviations: CMR = Cardiovascular magnetic resonance. EAT = Epicardial adipose tissue. IS = Infarct size. LAD = Left anterior descending. LV = Left ventricular. LVEDVI = Left ventricular end-diastolic volume index. LVEF = Left ventricular ejection fraction. LVESVI = Left ventricular end-systolic volume index. MVO = Microvascular obstruction. PCI = Percutaneous coronary intervention. SMD = Standardized mean differences. TIMI = Thrombolysis in Myocardial Infarction.

**Table 9 diagnostics-14-00368-t009:** Association of ∆EAT with 6-month LVEF. Multivariate linear regression analysis adjusted for parameters independently related to 6-month LVEF shown in Table 6.

	Unstandardized β Coefficient	*p*-Value
**Heart rate (bpm)**	−0.172	0.002
**TIMI flow grade before PCI (%)**	2.564	0.003
**Δ** **EAT (mL/m^2^)**	−0.027	0.747

Abbreviations: CMR = Cardiovascular magnetic resonance. EAT = Epicardial adipose tissue. LVEF = Left ventricular ejection fraction. PCI = Percutaneous coronary intervention. TIMI = Thrombolysis in Myocardial Infarction. ΔEAT = Variation of EAT from 1-week to 6-month CMR.

**Table 10 diagnostics-14-00368-t010:** Association of ∆EAT with 6-month LVEF. Multivariate lineal regression analyses adjusted for parameters independently related to 6-month LVEF shown in Table 6 in patients with non-extensive (A) and extensive (B) 1-week IS.

**A. *NON-EXTENSIVE IS***		
	**Unstandardized** ** β ** **Coefficient**	** *p* ** **-Value**
**Heart rate (bpm)**	−0.066	0.368
**TIMI flow grade before PCI (%)**	0.741	0.476
** Δ** **EAT (mL/m^2^)**	−0.101	0.435
**B. *EXTENSIVE IS***		
	**Unstandardized** ** β ** **Coefficient**	** *p* ** **-Value**
**Heart rate (bpm)**	−0.174	0.009
**TIMI flow grade before PCI (%)**	3.674	0.002
**Δ** **EAT (mL/m^2^)**	−0.347	0.003

Abbreviations: CMR = Cardiovascular magnetic resonance. EAT = Epicardial adipose tissue. IS = Infarct size. LVEF = Left ventricular ejection fraction. PCI = Percutaneous coronary intervention. TIMI = Thrombolysis in Myocardial Infarction. ΔEAT = Variation of EAT from 1-week to 6-month CMR.

## Data Availability

The data presented in this study are available on request from the corresponding author.

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
