# Peer review of "Impact of Epicardial Adipose Tissue on Infarct Size and Left Ventricular Systolic Function in Patients with Anterior ST-Segment Elevation Myocardial Infarction"

_diagnostics, 2024, doi:10.3390/diagnostics14040368_

Round 1

Reviewer 1 Report

Comments and Suggestions for Authors

The authors of the manuscript present results from an original prospective study on patients who underwent anterior myocardial infarction with ST-elevation (STEMI) and had reperfusion therapy, aiming to specify the impact of the volume of epicardial adipose tissue on infarction size and left ventricular systolic function. Assessment of these morphological and functional parameters was done by caradiac magnetic resonance, conducted at Week 1 and Month 6 after STEMI. The results of this study are improtant from clinical and scientific point of view, and would be interesting to many readers of "Diagnostics". The manuscript is generally well-structured.

I have the following recommendations to the authors, that could further improve the quality of their work:

1. The title should be modified, for example "Association between CMR-derived epicardial adipose tissue, infarct size and left ventricular systolic function in patients with anterior STEMI", or "Impact of  epicardial adipose tissue on infarct size and left ventricular systolic function in patients with anterior STEMI", or something like that. 

2. Lines 30-32: "ST-segment elevation myocardial infarction (STEMI) is one of the main causes of morbidity and mortality in our setting" - STEMI is actually among the leading causes for morbidity and premature death worldwide (and definitely among the European countries, North America and many parts of Asia), so "in our setting" should be corrected.

3. The authors state "Overall, EAT decreased at 6 months" - How would you explain/interpete that finding to the readers?

4. Do you have any data from your study about association between EAT volume and early/late post-STEMI mortality? If yes, it would be valuable to present these results. There data from other studies showing that EAT is strongly related to risk of rhythm disorders (mediated by cytokines produced by EAT adipocytes), including ventricular tachycardias, which are among the main causes for early and late post-STEMI lethal outcome. 

Comments on the Quality of English Language

English language is generally fine

Reviewer 2 Report

Comments and Suggestions for Authors

Gavara et al reported their work named "Dual role of CMR-derived epicardial adipose tissue on infarct size and residual systolic function post-STEMI' and concluded " and concluded "In STEMI patients, higher presence of EAT is associated with larger IS. Nevertheless, in patients with large infarctions, high EAT and greater subsequent EAT reduction are linked to more preserved LVEF in chronic phase. This dual and paradoxical effect of EAT fuels the need for further research in this field.". I have the following comments:

- Please avoid abbreviations in the manuscript title.

- Please report the interpretation of the "correlation coefficient" in the methods.

- For any plotted regression line, please add scatter plots as well to the same figure.

- Please try to add jitter plots to your boxplots.##

- Please specify the criteria for inclusion of variables into multivariate analyses.

- Please write "Unstandardized β coefficient" instead of "Unstandardized β coefficient" in all tables.

- Please define different "Killips classes", "TIMI flow grade" and "GRACE score" in the methods section or in the supplements.

- Minor language revision is essential.

- Please spell out any abbreviations in their first time eg HbA1c and ECG in Table 5.

- Tables 7, 9, and 10: Please delete "R=0.163; R2=0.027; Constant=58.741" and other similar values in other tables. 

- Please report Paired t-test P value for each group in Figure 7.

- Figure 8: Please consider fixing the legend site of Plot "A".

Round 2

Reviewer 2 Report

Comments and Suggestions for Authors

The authors addressed my prior comments and it is my pleasure to accept their work.

Author Response

Thank you very much for your comments.